# Physical contact transmission of Cucumber green mottle mosaic virus by *Myzus persicae*

Yu-Hua Qi[1,2], Yu-Juan He[1], Xin Wang[1,2], Chuan-Xi Zhang[1], Jian-Ping Chen[1,2], Gang Lu[1]*, Jun-Min Li[1]*

1 State Key Laboratory for Managing Biotic and Chemical Threats to the Quality and Safety of Agro-products, Key Laboratory of Biotechnology in Plant Protection of Ministry of Agriculture and Zhejiang Province, Institute of Plant Virology, Ningbo University, Ningbo, China, 2 College of Plant Protection, Fujian Agriculture and Forestry University, Fuzhou, China

* lugang@nbu.edu.cn (GL); lijunmin@nbu.edu.cn (JML)

**Data Availability Statement:** All relevant data are within the paper and its Supporting Information files.

## Abstract

Cucumber green mottle mosaic virus (CGMMV), a critical plant virus, has caused significant economic losses in cucurbit crops worldwide. It has not been proved that CGMMV can be transmitted by an insect vector. In this study, the physical contact transmission of CGMMV by *Myzus persicae* in *Nicotiana benthamiana* plants was confirmed under laboratory conditions. The acquisition rate increased with time, and most aphids acquired CGMMV at 72 h of the acquisition access period (AAP). Besides, the acquired CGMMV was retained in the aphids for about 12 h, which was efficiently transmitted back to the healthy *N. benthamiana* plants. More importantly, further experiments suggested that the transmission was mediated by physical contact rather than the specific interaction between insect vector and plant virus. The results obtained in our study contribute to the development of new control strategies for CGMMV in the field.

## Introduction

Plant viruses spread among their host plants in the field through a variety of ways, including insect vectors. To date, over 75% known plant viruses are transmitted in the field by insect vectors by three different modes, including non-persistent, semi-persistent and persistent [1]. Among them, the non-persistently transmitted viruses can survive in insect stylets for several minutes to several hours during the insect feeding processes. Meanwhile, the semi-persistently transmitted viruses can be retained in insect foreguts for a few hours to several days after insect acquisition. As for the persistently transmitted viruses, they first infect the epithelial cells of insect midguts or hindgut, then enter the hemolymph or other tissues, and move into the salivary glands. During this infection process, these viruses can survive inside the insect vectors for several days to the whole life [1–4]. In addition, the specific interactions between insect vector receptors and viral proteins, like glycoprotein and coat protein (CP), play an essential role in the acquisition and transmission of plant viruses in the field [5, 6].

Over 55% of the known insect-transmitted plant viruses are transmitted by different aphids [1]. *Myzus persicae*, which belongs to the family Aphididae, order Hemiptera, is an important agricultural pest, often causes serious damages to various food crops (e.g., crop plants in the

**Funding:** This work was supported by National Natural Science Foundation of China (U20A2036), Natural Science Foundation of Zhejiang Province (LQ20C140003), Natural Science Foundation of Ningbo City (2019A610404), Ningbo Science and Technology Innovation 2025 Major Project (2019B10004). This work was sponsored by K.C. Wong Magna Fund in Ningbo University. The funders had no role in study design, data collection and analysis, decision to publish, or preparation of the manuscript.

**Competing interests:** The authors have declared that no competing interests exist.

**Abbreviations:** AAP, Acquisition access period; CGMMV, Cucumber green mottle mosaic virus; COI, Cytochrome c oxidase subunit I; CP, Coat protein; IAP, Inoculation access period.

family Solanaceae, Cruciferae or Rosaceae) by sucking saps from their phloem tissues, and possess a great reproductive capacity [7, 8]. Moreover, *M. persicae* has been well known for its ability to transmit among over 100 plant viruses [9]. Some plant viruses, including Potato virus Y (PVY), Turnip mosaic virus (TuMV) and Cucumber mosaic virus (CMV), can be transmitted by *M. persicae* in a non-persistent manner [10–12]. Other viruses such as Turnip yellows virus (TuYV), Potato leafroll virus (PLRV), and Beet western yellows virus (BWYV) are transmitted by *M. persicae* in a persistent manner [13–15].

Cucumber green mottle mosaic virus (CGMMV) is a member in the genus *Tobamovirus*, family Virgaviridae. CGMMV often causes leaf mosaic and malformation, fruit mottling and plant stunting in numerous plant species, including those in the families Solanaceae, Cucurbitaceae and Cruciferae, leading to severe damages to the vegetable industry worldwide [16–18]. CGMMV is originally identified in cucumber plant in England, and it is now extensively distributed in Asia, Australia and America [19–26]. Similar to other tobamoviruses, CGMMV is mainly transmitted via contacts and seeds in the field [27]. Recent study has indicated that the beneficial pollinator insects contribute to the spread of CGMMV between plants during the foraging behavior of honey bees [28]. Nonetheless, it remains unclear about whether CGMMV can be transmitted by insects via physical contact or by the above three different transmission modes.

In this study, we determined that CGMMV was acquired by *M. persicae* directly fed on the CGMMV-infected plants, and that these infected aphids successfully transmitted CGMMV to other healthy plants. Further experiments confirmed that the transmission was mediated by physical contact rather than the specific interaction between aphids and CGMMV.

## Materials and methods

### Maintenance of insects and plants

The CGMMV-free aphid population was maintained on the *N. benthamiana* plants in a growth chamber under the conditions of 24˚C and the 14 h/10 h light/dark cycle. Thereafter, the aphids were transferred onto new plants at intervals of two weeks. The aphid species were confirmed by PCR amplification of the cytochrome c oxidase subunit I (COI) gene. The primers used for COI gene amplification are listed in S1 Table.

### Infectious clone of CGMMV by agrobacterium infiltration

The infectious clone of CGMMV was kindly presented by Professor Fei Yan from Ningbo University, China [26]. Thereafter, the plasmid pCB301-CGMMV was transformed into the *Argobacterium tumeficience* strain GV3101, followed by overnight propagation in the LB medium supplemented with 50 mg/μL kanamycin and 25 mg/μL rifampicin on a shaker at 28˚C. Afterwards, the Agrobacterium cells were pelleted by centrifugation at 5000 rpm for 5 min, re-suspended in the infiltration buffer, and diluted with the infiltration buffer to $OD_{600} = 1.0$. Later, the diluted Agrobacterium cultures were incubated for 2 h at 28˚C and subsequently infiltrated into the leaves of *N. benthamiana* plants. At 10 days after agrobacterium infiltration, the assayed plants were analyzed for CGMMV infection.

### CGMMV acquisition by *M. persicae*

About 100 non-viruliferous aphids were fed on the CGMMV-infected *N. benthamiana* plants. Afterwards, about 8 aphids were collected at 3, 6 and 72 h after feeding. Three biological replicates were set for each time point, with the CGMMV-infected or CGMMV-free aphids being the positive or negative controls, respectively.

## CGMMV retention in *M. persicae*

First of all, approximately 150 non-viruliferous aphids were fed on the CGMMV-infected *N. benthamiana* plants for 3 days in the insect-proof cage, and were later transferred onto the healthy plants. At 6, 12 or 72 h after feeding, various aphids on the healthy plants were collected from each group and checked for CGMMV retention. Three biological replicates were set for each time point. The same positive or negative controls of aphids were used as described above.

## CGMMV transmission to *N. benthamiana* by *M. persicae*

About 1000 non-viruliferous aphids were fed on the CGMMV-infected or CGMMV-free *N. benthamiana* plants for 72 h in different insect-proof cages, and were later transferred onto the new virus-free *N. benthamiana* plants. There were approximately 20 aphids in each plant. All aphids were removed after inoculation for 24 h, and the plants were left for symptom development. In addition, both the inoculated and systemic leaves from approximately 12 plants were collected for CGMMV detection by RT-PCR after 20 days. Three biological replicates were set in this experiment, with CGMMV-infected or CGMMV-free plants being the positive or negative controls, respectively.

## Western blotting

Proteins were extracted from plant leaf samples into the phosphate- buffered saline (PBS, pH 7.4). Afterwards, the extracted proteins were separated through 10% SDS-PAGE and later transferred onto the nitrocellulose membranes. Thereafter, the nitrocellulose membranes were blocked with 5% non-fat milk solution for 1 h, and incubated with the anti-CGMMV CP antibody diluted with PBS at 1:10,000 (v/v). Subsequently, the membranes were further incubated with goat anti-rabbit HRP-conjugated antibody diluted at 1:10,000 (v/v). Next, the membranes were additionally incubated with the mixture of SuperSigmal™ West Pico PlusLumino/Enhancer Solution and SuperSigmal™ West Pico Plus Stable Peroxide Solution (1:1) for 2 min to develop the detection signal. Finally, the Luminescent Image Analyzer AI680 (GE, Sweden) was employed to visualize the detection signals.

## Reverse Transcription Polymerase Chain Reaction (RT-PCR)

Total RNA was extracted from aphid or plant leaf samples with the TRIzol reagent (Invitrogen, USA). Later, concentrations of the extracted RNA were determined using the NanoDrop spectrophotometer (Thermo Scientific, USA). Then, 2 μL total RNA in 20 μL reaction solution was used to synthesize the first-strand cDNA using the Superscript III First-Strand Synthesis System (Vazyme) with random primers. Notably, PCR procedures with thirty cycles were conducted by adopting the specific primers of CGMMV CP gene, together with the primers of reference genes Actin (*M. persicae*) and UBC (*N. benthamiana*). The primers used for RT-PCR procedures are listed in S1 Table. Thereafter, the obtained PCR products were further confirmed by Sanger sequencing.

## CGMMV CP expression and purification

Using the following primer pair 28-CGMMV CP-F/28-CGMMV CP-R (S1 Table), the open reading frame (ORF) of CGMMV *CP* (with His tag) was constructed in the pET28a vector between the *Nde* I-*Bam* H I restriction sites. Then, plasmid pET28a-CGMMV CP was transferred and expressed in the *Escherichia coli* strain *BL21* (DE3). After induction with 0.6 mM isopropyl-β-D-thiogalactoside for 12 h at 16 ˚C, cells were pelleted by centrifugation at 5000

rpm for 10 min, re-suspended in PBS, and sonicated for 30 min. Then, the supernatants of sonicated cells were collected for protein purification. Later, the expressed recombinant protein CP-His was purified with the Ni-NTA Agarose (QIAGEN, Germany) following the manufacturer's instructions and then prepared for aphid feeding.

## CGMMV acquisition by *M. persicae* through membrane feeding

Approximately 100 non-viruliferous aphids were fed through the stretched parafilm membranes containing sap derived from the CGMMV-infected *N. benthamiana* leaves mixed with 20% (w/v) sucrose. Then, about 10–24 aphids were collected at 0.5, 3, 10 and 48 h post-feeding, respectively. On the other side, the CGMMV-infected *N. benthamiana* leaves were tiled in vitro on the petri dish coated with agar (wrapped inside the parafilm membrane), and 60 non-viruliferous aphids were fed on the infected leaves through the membrane. At 48 h post-feeding, 10 aphids were collected for virus detection. Consistently, the CGMMV-infected or CGMMV-free aphids served as the positive or negative controls, respectively.

## Evaluation of CGMMV transmission through membrane virus acquisition by *M. persicae*

After membrane virus acquisition described above, around 20 aphids were transferred onto each virus-free *N. benthamiana* plant. After 48 h of inoculation, aphids were removed and the plants were left for symptom development. After 20 days, both the inoculated and systemic plant leaves were harvested for CGMMV detection. Similarly, the CGMMV-infected or CGMMV-free plants were used as the positive or negative controls, respectively.

## Effect of CGMMV CP pre-treatment on the virus acquisition by *M. persicae*

Firstly, 60 non-viruliferous aphids were fed through the stretched membranes coated with the purified CGMMV CP or GFP protein (control) for 48 h, respectively. Later, the pre-fed aphids were transferred and fed on leaves of CGMMV-infected plants for 48 h. Thereafter, 12 aphids from each treatment were collected for virus detection. Accordingly, the CGMMV-infected or CGMMV-free aphids served as the positive or negative controls, respectively.

## Localization of CGMMV in *M. persicae*

The adult *M. persicae* was fed on CGMMV-infected *N. benthamiana* plants for 6 h. Afterwards, both the stylet and gut were collected from an individual *M. persicae* and fixed with the 4% paraformaldehyde solution (Thermo Fisher Scientific, USA) overnight at 4°C. Subsequently, the fixed stylets and guts were washed with 0.01 M PBS (pH 7.4) thrice, exposed to the 2% Triton X-100 solution for 30 min, and incubated with the CGMMV CP specific antibody diluted at 1:200 (v/v) for 1 h. Later, the stylets were washed with PBS thrice, incubated with the TRITC-conjugated goat anti-rabbit IgG diluted at 1:200 (v/v) for another 1 h, and then washed with PBS thrice again. Accordingly, the guts were washed with PBS thrice, incubated with the TRITC- and FITC-conjugated goat anti-rabbit IgG diluted at 1:200 (v/v) for additional 1 h, and then washed with PBS thrice again. Thereafter, the labeled stylets and guts were added into the anti-fluorescence attenuation solution (Solarbio, China) on the microscope slides, and observed under the inverted Nikon ECLIPSE Ni-E confocal microscope (Nikon, China). The excitation wavelength for TRITC was set at 50 μm. Approximately 100 individual aphids were used in this experiment.

## Results

### Acquisition of CGMMV by *M. persicae*

To determine whether *M. persicae* acquired the CGMMV virions through feeding, CGMMV-infected *N. benthamiana* plants were used as the inoculation source via agroinfiltration. At 10 days after inoculation, curling symptoms were observed in systemic leaves of *N. benthamiana*. Moreover, CGMMV infection in these systemic leaves was confirmed by both RT-PCR and Western blotting (Fig 1). Thereafter, *M. persicae* was fed on the systemic leaves of CGMMV-infected or non-infected (control) plants. At 3, 6 and 72 h of the acquisition access period (AAP), aphids were collected for CGMMV detection. According to the results of RT-PCR analysis, about 25% aphids fed on the CGMMV-infected plants acquired CGMMV at 3 h of AAP (Fig 2A and 2D, Table 1). At 6 h of AAP, around 67% aphids fed on the CGMMV-infected plants acquired CGMMV (Fig 2B and 2D, Table 1). At 72 h of AAP, most aphids acquired CGMMV (Fig 2C and 2D, Table 1). Noteworthily, the amount of CGMMV detected in individual aphids varied significantly (Fig 2A). By contrast, no CGMMV was detected in aphids fed on the non-infected plants as expected.

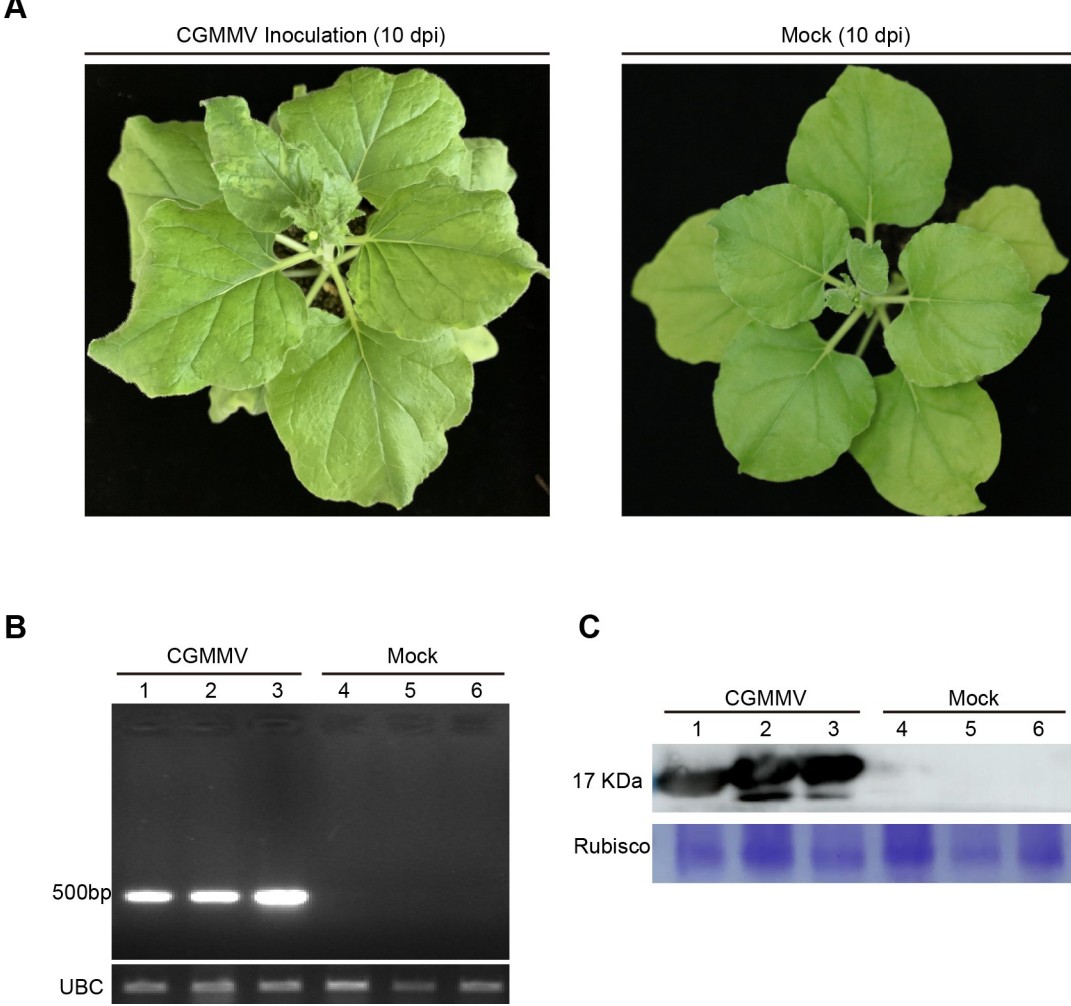

**Fig 1. Agroinfiltration of CGMMV to the *N. benthamiana* plants.** (A) Symptoms of CGMMV/mock-inoculated *N. benthamiana* at 10 dpi. (B) RT-PCR analysis on the CGMMV/mock-inoculated *N. benthamiana* plants. (C) Western blotting analysis on the CGMMV/mock-inoculated *N. benthamiana* plants. dpi: days post inoculation.

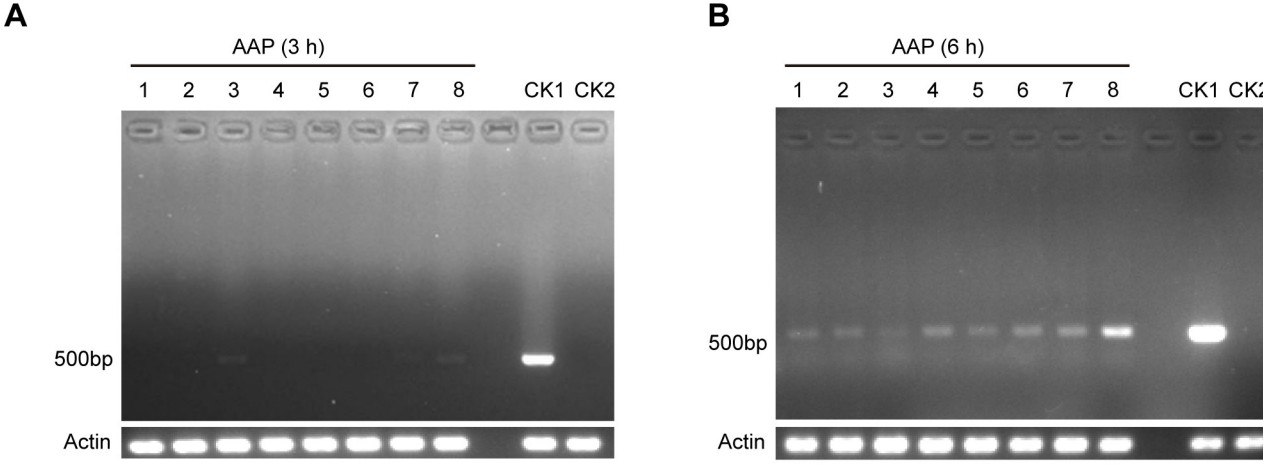

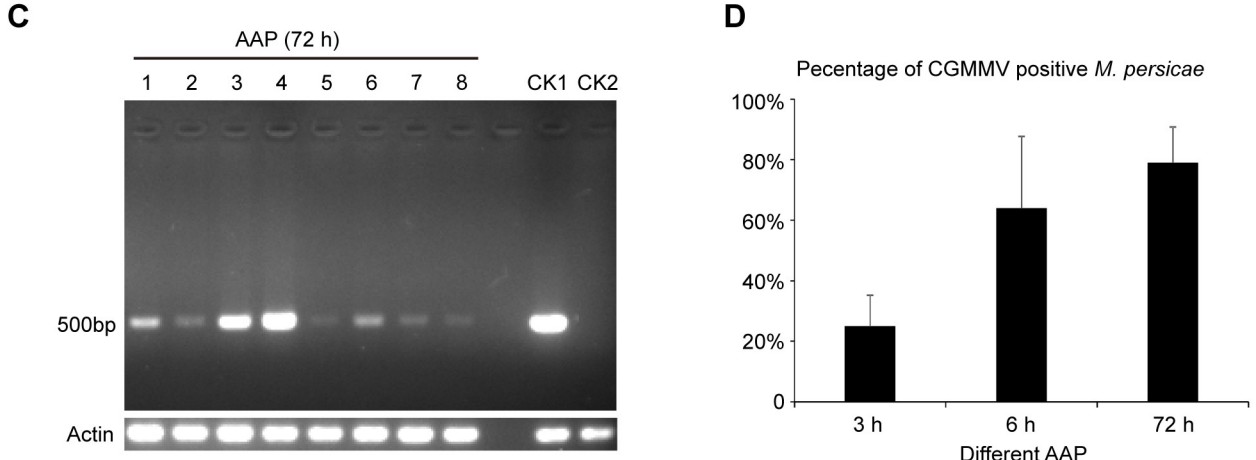

**Fig 2. Acquisition of CGMMV from CGMMV-infected *N. benthamiana* plants by *M. persicae.*** RT-PCR analysis on CGMMV in *M. persicae* fed on the CGMMV-infected *N. benthamiana* plants for 3 h (A), 6 h (B) and 72 h (C). Three biological replicates were set for each treatment. The CGMMV acquisition rate (Table 1) was calculated by *t*-tests (D). CK1, positive control (aphids fed on CGMMV-infected *N. benthamiana* plants for 7 days); CK2, negative control (aphids fed on CGMMV-free *N. benthamiana* plants). AAP: Acquisition access period.

**Table 1. CGMMV acquisition by *M. persicae.***

| Different AAP* (h) | Replication1 | Replication2 | Replication3 |
|---|---|---|---|
| | Positive aphids/Total aphids (Percentage) | Positive aphids/Total aphids (Percentage) | Positive aphids/Total aphids (Percentage) |
| 3 | 1/8 (12%) | 3/8 (37%) | 2/8 (25%) |
| 6 | 3/6 (50%) | 8/8 (100%) | 3/6 (50%) |
| 72 | 6/8 (75%) | 6/8 (75%) | 8/8 (100%) |

*AAP: Acquisition access period.

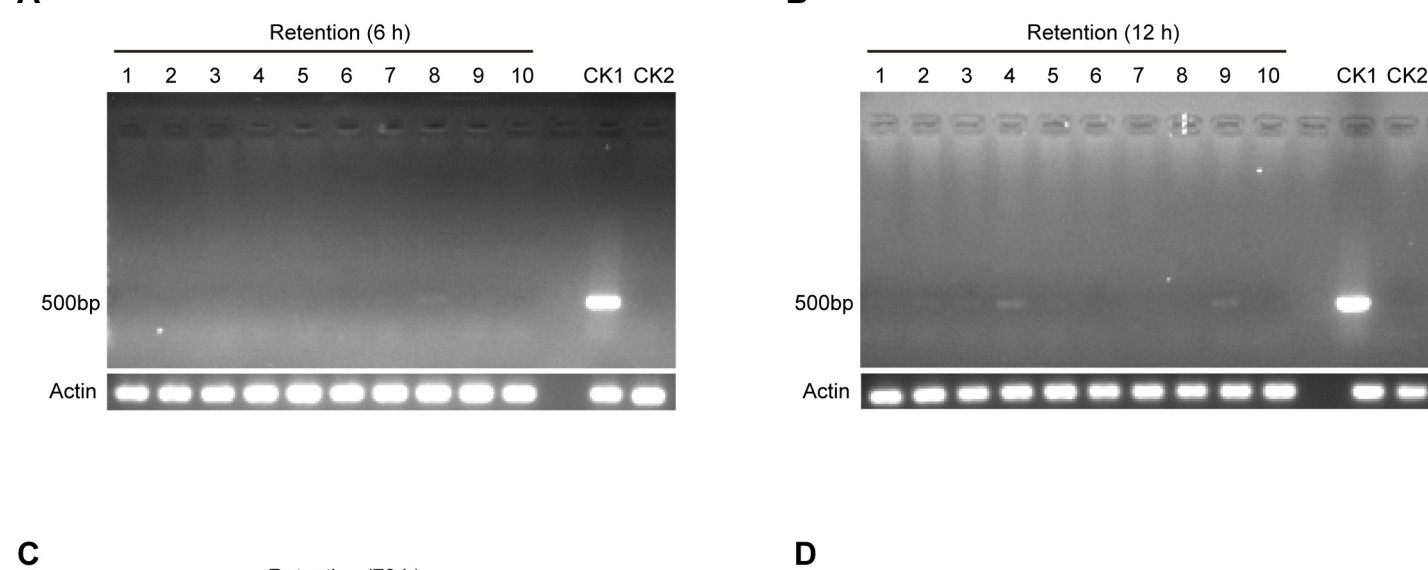

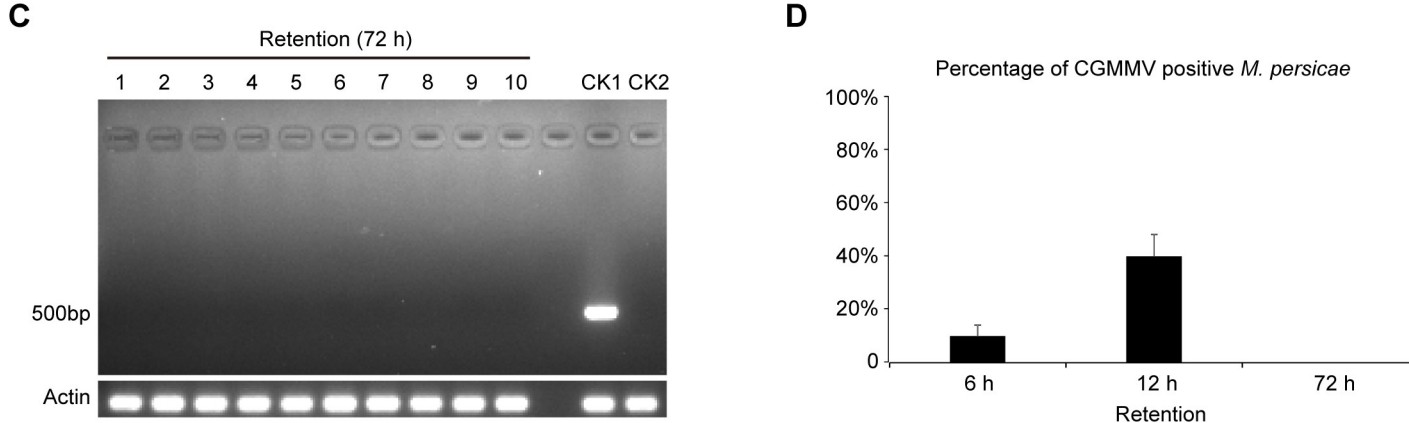

**Fig 3. Retention of CGMMV in *M. persicae*.** RT-PCR analysis on CGMMV retention in *M. persicae* fed on healthy plants at 6 h (A), 12 h (B) and 72h (C). Three biological replicates were set for each treatment. The CGMMV retention rate (Table 2) was calculated by *t*-tests (D). CK1, positive control (aphids fed on CGMMV-infected *N. benthamiana* plants for 7 days); CK2, negative control (aphids fed on CGMMV-free *N. benthamiana* plants).

## Retention time of CGMMV in *M. persicae*

At 6 h of AAP, aphids in CGMMV-infected plants were transferred onto the non-infected *N. benthamiana*. At 6, 12 and 72 h after feeding on the non-infected plants, aphids were tested for CGMMV retention. As a result, at 6 and 12 h after feeding on the healthy plants, the CGMMV retention rates in aphids were about 10% and 40%, respectively (Fig 3A, 3B and 3D, Table 2). At 72 h after feeding on the healthy plants, no CGMMV retention was detected in the sampled aphids (Fig 3C and 3D, Table 2).

**Table 2. Retention of CGMMV in *M. persicae*.**

| Retention time (h) | Replication1 | Replication2 | Replication3 |
|---|---|---|---|
| | Positive aphids/Total aphids (Percentage) | Positive aphids/Total aphids (Percentage) | Positive aphids/Total aphids (Percentage) |
| 6 | 3/20 (15%) | 1/10 (10%) | 1/20 (5%) |
| 12 | 5/10 (50%) | 3/10 (30%) | 4/10 (40%) |
| 72 | 0/6 (0%) | 0/7 (0%) | 0/8 (0%) |

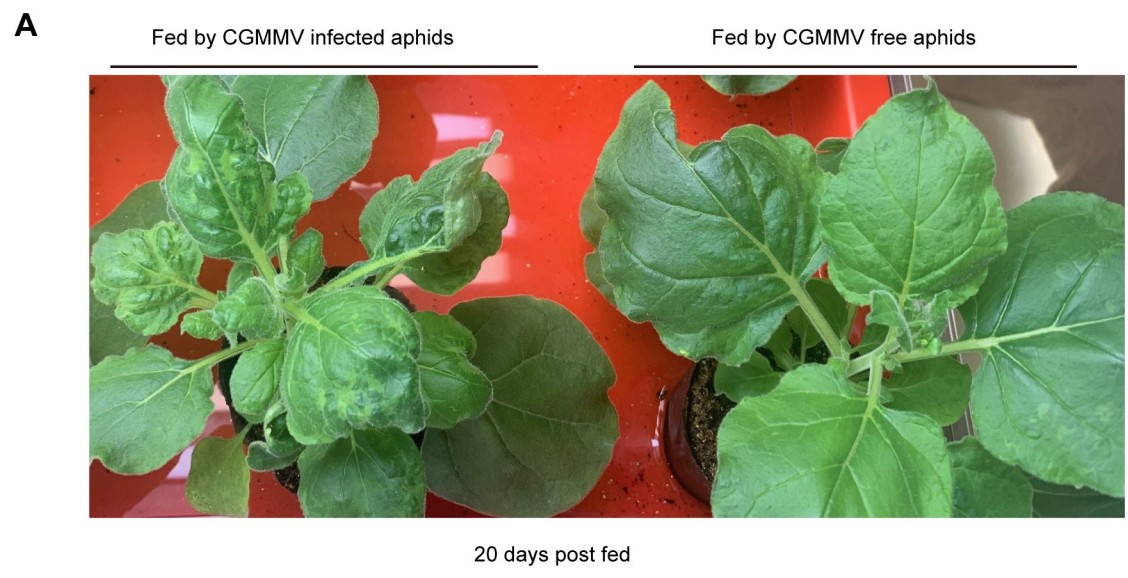

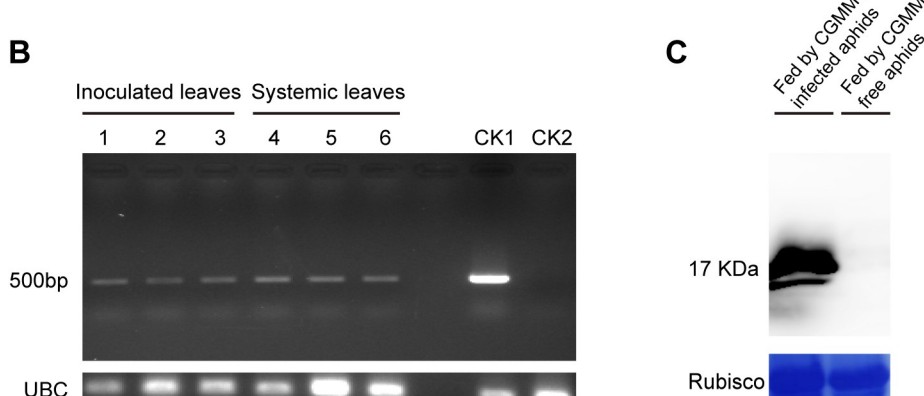

**Fig 4. Transmission of CGMMV to *N. benthamiana* by *M. persicae*.** (A) Symptom development of *N. benthamiana* plants fed by CGMMV-infected and CGMMV-free *M. persicae* at 20 day post fed. (B) RT-PCR analysis on CGMMV in systemic leaves and leaves inoculated with the CGMMV-infected *M. persicae*. CK1, positive control (CGMMV-infected *N. benthamiana* plants); CK2, negative control (CGMMV-free *N. benthamiana* plants). dpi: days post inoculation. (C) Western blotting analysis on the systemic leaves of *N. benthamiana* plants fed by CGMMV-infected and CGMMV-free *M. persicae*.

### Successful transmission of CGMMV to *N. benthamiana* by *M. persicae*

To investigate whether *M. persicae* transmitted CGMMV to the healthy *N. benthamiana* plants, *M. persicae* was fed on the CGMMV-infected *N. benthamiana* plants for 6 h and then transferred into the healthy *N. benthamiana* plants at 24 h of inoculation access period (IAP). Thereafter, the inoculated *N. benthamiana* plants were transferred to an insect-free growth chamber. At 20 days later, the CGMMV-like disease symptoms were observed in systemic leaves of most aphid-inoculated *N. benthamiana* plants (Fig 4A). Besides, as indicated by RT-PCR and Western blotting analysis, CGMMV accumulated in both the aphid-inoculated and systemic leaves of the assayed plants (Fig 4B and 4C), and the CGMMV transmission rate was about 80% (Table 3).

### Successful CGMMV acquisition by *M. persicae* fed through the double-parafilm membrane

The above results suggested that *M. persicae* successfully acquired and transmitted CGMMV in the *N. benthamiana* plants. On this basis, it is interesting to further explore whether the transmission is mediated by the direct interaction between aphids and virus, or merely by the physical contact reported previously for CGMMV and other tobamoviruses [27]. To investigate the possibility of transmission mediated by physical contact, *M. persicae* was fed through the two-layer stretched parafilm containing sap derived from the CGMMV-infected leaves mixed with 20% (w/v) sucrose, and then insects were collected at four different AAPs (0.5, 3, 10 and 48 h, respectively). As revealed by the RT-PCR results, the percentage of viruliferous aphids steadily increased as the AAP extended, and all the sampled aphids were CGMMV-positive at 48 h of APP (Table 4). Simultaneously, the other group of aphids was fed on the CGMMV-infected *N. benthamiana* leaves wrapped inside the parafilm membrane for 48 h. Consequently, the virus acquisition percentage was 40%, lower than that of aphids fed on leaf sap (Table 4).

### CGMMV acquired by *M. persicae* through membrane feeding was not transmitted into the healthy plants

To evaluate whether the membrane-acquired CGMMV was transmitted to the healthy plants by *M. persicae*, we transferred the treated aphids to the virus-free plants for 48 h to achieve virus inoculation. Thereafter, the plants were left for symptom development. As a result, no symptom was observed in the inoculated plants after 20 days. Besides, the results of RT-PCR analysis indicated that no CGMMV was detected in the aphids-inoculated plants, no matter whether the aphids acquired viruses through the parafilm-packaged leaf sap or plant leaves (S2 Table). These results demonstrated that aphids acquired CGMMV by feeding through the parafilm membrane, but it did not transmit the acquired virus to the healthy plants.

### Pre-feeding of CGMMV CP showed no effect on virus acquisition in *M. persicae*

To further determine whether CGMMV CP interacted with the aphid receptors that might facilitate virus transmission, aphids were pre-fed with the purified CGMMV CP or GFP protein (control) for 48 h. Thereafter, the ability to acquire CGMMV from the infected plants was evaluated. After 48 h of acquisition, results of RT-PCR analysis revealed no difference in virus acquisition between the CP pre-treated aphids and controls, and all the sampled aphids successfully acquired CGMMV (S3 Table), indicating that CGMMV CP had no effect on virus acquisition by the aphids.

### CGMMV was not detected in the stylets or midguts of *M. persicae*

Given the successful acquisition of CGMMV by *M. persicae*, CGMMV CP antibody was used for the immunolabeling of stylets and midguts to further explore the virion localization in the insects at 6 h of AAP. Our results indicated that none of the sampled aphid individuals exhibited clear CGMMV-labeled red florescence signal in the insect stylets or guts under the

**Table 3. Transmission of CGMMV to *N. benthamiana* by *M. persicae*.**

|  | Replication1 | Replication2 | Replication3 |
|---|---|---|---|
| Positive plants/Total plants (Percentage) | 8/12 (66.7%) | 8/11 (72.7%) | 11/13 (84.6%) |

**Table 4. CGMMV acquisition by *M. persicae* through parafilm membrane feeding.**

| | Parafilm-packaged leaves sap | | | | Parafilm-packaged leaves |
|---|---|---|---|---|---|
| Different AAP* (h) | 0.5 | 3 | 10 | 48 | 48 |
| Positive aphids/Total aphids (Percentage) | 4/24 (16%) | 3/12 (25%) | 4/12 (33%) | 10/10 (100%) | 4/10 (40%) |

*AAP: Acquisition access period

confocal microscope (S1A and S1C Fig), consistent with the results observed in controls (aphids fed on virus-free *N. benthamiana* plants) (S1B and S1D Fig).

## Discussion

As reported in numerous studies, tobamoviruses in the field mainly spread through seeds and/or physical contacts [29–32]. In addition, honey bees, the beneficial insect, are also suggested to contribute to CGMMV transmission in the field, which may be achieved through physical contact [28]. In this study, we discovered that CGMMV was successfully acquired, retained and transmitted in *N. benthamiana* plants by *M. persicae*. Moreover, further evidence convincingly demonstrated that *M. persicae* (an important plant virus vector) mainly transmitted CGMMV through physical contact.

*M. persicae* can transmit over 100 plant viruses, mostly the non-persistent stylet-borne viruses [2, 9]. Generally, non-persistently transmitted plant viruses can only be retained by their vectors for a very short period of time (several minutes), thereby resulting in the short-run virus transmission in the field. However, these viruses are often the effective transmitters, which is mainly ascribed to their short AAP and IAP (ranging from seconds to minutes) [2]. For semi-persistent foregut-borne viruses, their AAP and IAP range from minutes to hours, while their retention time lasts for several hours [2]. Our results showed that *M. persicae* acquired CGMMV from the infected plants within about 3–6 h (Fig 2, Table 1), and the CGMMV retention time was over 12 h (Fig 3, Table 2), suggesting the semi-persistent transmission. Unexpectedly, the percentage of CGMMV-retaining aphids increased after they were transferred to the healthy plants for 12 h compared with that for 6 h (Fig 3, Table 2). One possible reason was that the originally acquired viral RNA amounts were different between these sets of aphids. Nevertheless, the acquired CGMMV was efficiently transmitted to the healthy plants by *M. persicae* (Fig 4, Table 3).

At present, there is no distinct evidence indicating that CGMMV can be transmitted by insects through the interaction between virus and vector [33]. Therefore, this study further explored the details of CGMMV transmission by *M. persicae*. A majority of plant virus vectors are phloem-limited hemipteran insects, and the interaction between virus and insect is initiated through phloem feeding [2]. To address whether the physical contact between aphids and CGMMV-infected leaves contributed to virus acquisition and transmission, we fed aphids through the parafilm-packaged leaf sap or plant leaves to acquire CGMMV. Interestingly, our results indicated that although aphids successfully acquired CGMMV through membrane feeding, they failed to transmit the CGMMV to the healthy plants (Table 4 and S2 Table). This result strongly indicated that CGMMV transmission might be mediated by physical contact rather than by the classical vector transmission modes of plant viruses. In addition, previous studies have suggested that virions can be acquired by insect vectors through membrane feeding, and Alfalfa mosaic alfamovirus [34], rather than Turnip mosaic potyvirus, can be successfully transmitted to the healthy plants [35]. The above phenomenon may be ascribed to the limited quantity of virions obtained, which is below the minimum viral amount threshold for success transmission, as described previously [36].

The specific interactions between viral proteins and insect vector receptors play important roles in the successful transmission of plant viruses [6]. Another evidence for the non-specific vector transmission of CGMMV was that, there was no significant difference in virus acquisition between aphids pre-fed with the purified CGMMV CP and the controls, indicating that CP had no effect on CGMMV transmission in *M. persicae*. Moreover, no CGMMV-labeled red florescence signals were detected in the stylets or guts of aphids fed on the infected plants, suggesting that there might be no specific interactions between viral proteins and insect receptors. Given that CGMMV was detected in the aphids by RT-PCR after virus acquisition (Fig 2), the absence of CGMMV-labeled signals might be ascribed to the fact that virions were rinsed away during the experimental process because of non-specific binding.

Collectively, our results suggest that CGMMV can be successfully acquired and transmitted by *M. persicae*, and the transmission is mediated by physical contact but not the specific interaction between the vector and the virus. Also, it is interesting to further investigate whether CGMMV or other tobamoviruses can also be physically transmitted by insect in the field. Findings in this study contribute to the development of effective management strategies for CGMMV control in the economically important food crops.

## Supporting information

**S1 Fig. Detection of CGMMV virions in the guts and stylets of *M. persicae*.** (A) Guts of *M. persicae* fed on CGMMV-infected plants. (B) Guts of *M. persicae* fed on CGMMV-free plants. (C) Stylets of *M. persicae* fed on CGMMV-infected plants. (D) Stylets of *M. persicae* fed on CGMMV-free plants. Red fluorescence signals indicate the localization of CGMMV in the guts or stylets of aphids. Green fluorescence signals indicate the actin gene in the guts of aphids. Scale bars, 100 μm.
(TIF)

**S1 Table. Primers used in this study.**
(DOCX)

**S2 Table. Transmission of CGMMV after viral acquisition through parafilm membrane by *M. persicae*.**
(DOCX)

**S3 Table. Effect of CGMMV CP pre-treatment on virus acquisition of *M. persicae*.**
(DOCX)

**S1 Raw images. The original blot and gel images in each figure.** Molecular weight markers were indicated in the left of each raw image, and each lane not included in the figure was marked with an "X" above the lane.
(PDF)

## Acknowledgments

We thank Professor Xin-Shun Ding for his valuable and constructive suggestions for improving the manuscript.

## Author Contributions

**Conceptualization:** Chuan-Xi Zhang, Gang Lu, Jun-Min Li.

**Data curation:** Yu-Hua Qi, Xin Wang, Gang Lu.

**Formal analysis:** Yu-Hua Qi, Xin Wang, Gang Lu.

**Funding acquisition:** Jian-Ping Chen, Gang Lu.

**Investigation:** Yu-Hua Qi, Yu-Juan He, Xin Wang.

**Project administration:** Chuan-Xi Zhang, Jian-Ping Chen, Gang Lu.

**Supervision:** Chuan-Xi Zhang, Jian-Ping Chen, Gang Lu, Jun-Min Li.

**Validation:** Yu-Juan He, Xin Wang, Chuan-Xi Zhang, Jian-Ping Chen, Jun-Min Li.

**Visualization:** Yu-Hua Qi, Yu-Juan He, Xin Wang, Jun-Min Li.

**Writing – original draft:** Yu-Hua Qi.

**Writing – review & editing:** Gang Lu, Jun-Min Li.

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
