## [Decision Letter · Decision Letter 0]

14 May 2021

PONE-D-21-10891

Mechanical transmission of Cucumber green mottle mosaic virus by Myzus persicae

PLOS ONE

Dear Dr. Lu,

Thank you for submitting your manuscript to PLOS ONE. After careful consideration, we feel that it has merit but does not fully meet PLOS ONE’s publication criteria as it currently stands. Therefore, we invite you to submit a revised version of the manuscript that addresses the points raised during the review process.

We look forward to receiving your revised manuscript.

Kind regards,

Sek-Man Wong

Academic Editor

PLOS ONE

Journal Requirements:

5. Please include your tables as part of your main manuscript and remove the individual files. Please note that supplementary tables (should remain/ be uploaded) as separate "supporting information" files.

[his work was supported by National Natural Science Foundation of China (U20A2036), Natural Science Foundation of Zhejiang Province (LQ20C140003), Natural Science Foundation of Ningbo City (2019A610404), Ningbo Science and Technology Innovation 2025 Major Project (2019B10004). This work was sponsored by K.C.Wong Magna Fund in Ningbo University.]

 [NO. The funders had no role in study design, data collection and analysis, decision to publish, or preparation of the manuscript.]

Reviewers' comments:

Reviewer's Responses to Questions

**Comments to the Author**

1. Is the manuscript technically sound, and do the data support the conclusions?

Reviewer #1: Yes

Reviewer #2: Partly

2. Has the statistical analysis been performed appropriately and rigorously? 

Reviewer #1: Yes

Reviewer #2: Yes

3. Have the authors made all data underlying the findings in their manuscript fully available?

Reviewer #1: Yes

Reviewer #2: Yes

4. Is the manuscript presented in an intelligible fashion and written in standard English?

Reviewer #1: No

Reviewer #2: Yes

5. Review Comments to the Author

Reviewer #1: The authors did lots of experiments to test the transmission of CGMMV by aphids. Though lots of data supported the idea that CGMMV can be transmitted by aphids, I donot agree the mechanical transimission by this insect. From these data, I suggest the authors to revise the paper to say that CGMMV can be transimitted by M. persicae by physical contaction. The aphids might get virions on the surface of their body.

L66. As more and more geminiviruses are found, the percent of aphid-borne viruses decreased.

The family names for animal and plants are not required in italic.

L74-79 Following the new writing style, the name of a virus is not in italic. Only describing the taxonomy of a virus species, the name must be italic.

How many cycles for RT-PCR in Figure 2 and 3? The bands for AAP and retension are faint. Usually, such faint bands are false positive amplification.

How many days it requires for mechanical transmission of CGMMV on N. benthamiana plants for systemic symptom development?

Figure 4A. not CGMMV infected aphids. Western blot or ELISA is required to confirm the infection of CGMMV on aphid-mechnically transmitted plants.

How to explain that successful CGMMV acquisition by M. persicae fed through the double-parafile membrane, but CGMMV was not detected in the stylets or midguts of M. persicae?

L400-401 I donot agree that CGMMV can be successfully acquired by M. persicae. From the results in this study, I think that M. persicae contacts virions in low chance.

Reviewer #2: In this manuscript “Mechanical transmission of Cucumber green mottle mosaic virus by Myzus persicae”, Qi et al. found that the M. persicae can acquire the CGMMV virions from virus-infected N. benthamiana and double-parafilm membrane containing leaf sap derived from the virus-infected N. benthamiana, and the virus can be retained by aphid for about 12 h after feeding on the N. benthamiana. Although the virus can be transmitted into healthy plant by aphids, the aphid feeding on the double-parafilm membrane cannot. In addition, the virus particles were not detected in the aphids. These results suggest that the M. persicae transmitted the virus by mechanical contact.

Many references in the introduction section are out of date. For example, Line 56-58, there are several other tissues have been identified as the initial infection site for different viruses, not only the midgut. I suggest that new research progresses should be added.

Line 35, 256, “CGGMV” should be “CGMMV”.

Figure 1A, the growth stage of the two plant are obviously different. Mock plant seems not to be agroinfiltrated and much younger than the left N. benthamiana plant.

6. PLOS authors have the option to publish the peer review history of their article (what does this mean?). If published, this will include your full peer review and any attached files.

Reviewer #1: No

Reviewer #2: No

---

## [Author Response · Author response to Decision Letter 0]

21 May 2021

Comments to the Author

1. Is the manuscript technically sound, and do the data support the conclusions?

Reviewer #1: Yes

Reviewer #2: Partly

Response: We appreciate the valuable comments and suggestions for the two Reviewers. We have revised the manuscripts accordingly.

2. Has the statistical analysis been performed appropriately and rigorously?

Reviewer #1: Yes

Reviewer #2: Yes

Response: Thanks for the positive comments.

3. Have the authors made all data underlying the findings in their manuscript fully available?

Reviewer #1: Yes

Reviewer #2: Yes

Response: Thanks for the positive comments.

4. Is the manuscript presented in an intelligible fashion and written in standard English?

Reviewer #1: No

Reviewer #2: Yes

Response: The manuscript has been carefully revised by a native English speaker for improvement of English writing.

5. Review Comments to the Author

Reviewer #1: The authors did lots of experiments to test the transmission of CGMMV by aphids. Though lots of data supported the idea that CGMMV can be transmitted by aphids, I donot agree the mechanical transimission by this insect. From these data, I suggest the authors to revise the paper to say that CGMMV can be transimitted by M. persicae by physical contaction. The aphids might get virions on the surface of their body.

Response: Thank you very much for your valuable comments and we do agree with your suggestion. We have checked and replaced the word “mechanical” with “physical contact transmission” as suggested. Please see in the revised manuscript. 

L66. As more and more geminiviruses are found, the percent of aphid-borne viruses decreased.

The family names for animal and plants are not required in italic.

L74-79 Following the new writing style, the name of a virus is not in italic. Only describing the taxonomy of a virus species, the name must be italic.

Response: Thank you for the suggestions. The change has been made in the revised manuscript (Please see Line68-85).

How many cycles for RT-PCR in Figure 2 and 3? The bands for AAP and retension are faint. Usually, such faint bands are false positive amplification.

Response: We are sorry for the confusion. Thirty cycles were set for the RT-PCR assay (Please see Line 173). To avoid false positives, we performed a negative control (CK2) that the aphids fed on CGMMV-free N. benthamiana plants in each experiment (Please see Figure 2 and 3). 

How many days it requires for mechanical transmission of CGMMV on N. benthamiana plants for systemic symptom development?

Response: It takes 20 days for the systemic symptom post aphid fed (Please see revised Figure 4). 

Figure 4A.not CGMMV infected aphids. Western blot or ELISA is required to confirm the infection of CGMMV on aphid-mechnically transmitted plants.

Response: Thank you for the suggestions. The Western blot assay has been added as suggested (Please see revised Figure 4). 

How to explain that successful CGMMV acquisition by M. persicae fed through the double-parafile membrane, but CGMMV was not detected in the stylets or midguts of M. persicae?

Response: The first possibility might be that the low amount of CGMMV virion was hardly detected by immune-labeling. And another possibility is that virion was rinsed away during the experimental process because of the non-specific binding property (Please see Line398-400).

L400-401 I donot agree that CGMMV can be successfully acquired by M. persicae. From the results in this study, I think that M. persicae contacts virions in low chance.

Response: Thank you for your valuable comment and we do agree with your suggestion that M. persicae contacts virions in low chance. 

Reviewer #2: In this manuscript “Mechanical transmission of Cucumber green mottle mosaic virus by Myzuspersicae”, Qi et al. found that the M. persicae can acquire the CGMMV virions from virus-infected N. benthamiana and double-parafilm membrane containing leaf sap derived from the virus-infected N. benthamiana, and the virus can be retained by aphid for about 12 h after feeding on the N. benthamiana. Although the virus can be transmitted into healthy plant by aphids, the aphid feeding on the double-parafilm membrane cannot. In addition, the virus particles were not detected in the aphids. These results suggest that the M. persicae transmitted the virus by mechanical contact.

Response: Thank you very much for your valuable comments.

Many references in the introduction section are out of date. For example, Line 56-58, there are several other tissues have been identified as the initial infection site for different viruses, not only the midgut. I suggest that new research progresses should be added.

Response: We do agree with your suggestion, and new references are added (Please see Line 57 in the revised manuscript).

Line 35, 256, “CGGMV” should be “CGMMV”.

Response: We are sorry for the mistake. We have revised it (Please see Line 35, 257 in the revised manuscript.

Figure 1A, the growth stage of the two plant are obviously different. Mock plant seems not to be agroinfiltrated and much younger than the left N. benthamiana plant.

Response: We do agree with your comments, and mock plant is replaced (Please see Figure 1A).

6. PLOS authors have the option to publish the peer review history of their article (what does this mean?). If published, this will include your full peer review and any attached files.

Do you want your identity to be public for this peer review? For information about this choice, including consent withdrawal, please see our Privacy Policy.

Reviewer #1: No

Reviewer #2: No

Response: Thanks for the positive comments.

---

## [Editor Report · Decision Letter 1]

25 May 2021

Physical contact transmission of Cucumber green mottle mosaic virus by Myzus persicae

PONE-D-21-10891R1

Dear Dr. Lu,

We’re pleased to inform you that your manuscript has been judged scientifically suitable for publication and will be formally accepted for publication once it meets all outstanding technical requirements.

Kind regards,

Sek-Man Wong

Academic Editor

PLOS ONE
---

## [Editor Report · Acceptance letter]

15 Jun 2021

PONE-D-21-10891R1 

Physical contact transmission of Cucumber green mottle mosaic virus by*Myzus persicae*

Dear Dr. Lu:

I'm pleased to inform you that your manuscript has been deemed suitable for publication in PLOS ONE. Congratulations! Your manuscript is now with our production department. 

Kind regards, 

on behalf of

Dr Sek-Man Wong 

Academic Editor

PLOS ONE